# Paper-Strip-Based Sensors for H_2_S Detection: A Proof-of-Principle Study

**DOI:** 10.3390/s22093173

**Published:** 2022-04-21

**Authors:** Maria Strianese, Viktoriia Vykhovanets, Naym Blal, Daniela Guarnieri, Alessandro Landi, Marina Lamberti, Andrea Peluso, Claudio Pellecchia

**Affiliations:** Dipartimento di Chimica e Biologia “Adolfo Zambelli”, Università degli Studi di Salerno, Via Giovanni Paolo II, 132, 84084 Fisciano, Italy; v.vykhovanets@studenti.unisa.it (V.V.); n.blal@studenti.unisa.it (N.B.); dguarnieri@unisa.it (D.G.); alelandi1@unisa.it (A.L.); mlamberti@unisa.it (M.L.); apeluso@unisa.it (A.P.)

**Keywords:** sensors, fluorescence, H_2_S

## Abstract

In this work, the authors explored the interaction of a suite of fluorescent zinc complexes with H_2_S. The authors provide evidence that HS^−^ binds the zinc center of all the complexes under investigation, allowing them to possibly function as sensors by a ‘coordinative-based’ approach. Naked-eye color changes occur when treating the systems with HS^−^, so the fluorescence responses are modulated by the presence of HS^−^, which has been related to a change in the energy level and coupling of excited states through a computational study. The results show the potential of the systems to function as HS^−^/H_2_S colorimetric and fluorescent sensors. Paper-strip-based sensing experiments foresee the potential of using this family of complexes as chemosensors of HS^−^ in more complex biological fluids.

## 1. Introduction

At the end of the 2000s, hydrogen sulfide (H_2_S) became the third small molecule that can exert both toxic and beneficial roles depending on its concentration, joining nitric oxide (nitrogen monoxide) and carbon monoxide and thus entering the family of gasotransmitters. In spite of the discovery that it is an endogenously-produced biomolecule and of its impressive therapeutic potential, the underlying mechanisms for its beneficial effects are not completely understood yet [1]. The past few decades have witnessed significant development of chemical tools for detection and delivery of H_2_S and related reactive sulfur species. Of such species, particular attention has focused on the development of biomimetic metal complexes, which may constitute major targets for gasotransmitters. Indeed, different than CO and NO, for which significant work focusing on their coordination chemistry has been published, H_2_S reactivity with bioinorganic scaffolds is still limited [2,3].

On the other hand, zinc is a biocompatible element that exists as the second most abundant transition metal ion and an indispensable trace element in the human body. d^10^ Zn(II) complexes not only exhibit large Stokes shifts and good photon stability but also possess strong emission and low cytotoxicity. With these considerations in mind, in the framework of our ongoing studies aiming at further understanding the coordination chemistry of H_2_S/HS^−^ with bioinorganic targets [4,5,6,7], the authors focused on zinc tetradentate Schiff-based complexes and proved that these systems function as viable scaffolds for isolating and characterizing hydrosulfido species and as very efficient HS^−^ sensing constructs via a ‘coordinative-based’ mechanism [8,9,10,11]. Since it is well acknowledged in the literature that ligand-dependent fluorescence represents a key feature of Zn-Salen complexes, and that it can be fine-tuned by a proper design of the ligands’ electronic states [12,13], the authors decided to focus on zinc salen-type complexes. In particular, it has been reported that modulation of the photophysical properties of the ZnSalen complexes is both dependent on the electronic states of the diamine moiety bridging the two nitrogens, which chelate the zinc on the substituents on the salycilaldehyde units [14]. In the current work, the authors decided to use a thiofuran-based diamine moiety as a bridging unit between the two Zn-chelating nitrogens (see Figure 1) to investigate its possible effects on the zinc hydrosulfido stabilization. The authors also wanted to explore whether different combinations of groups on the ligand structure (e.g., bridging diamine and substituents on the salicylaldehyde units with different electron-withdrawing and electron-donating abilities) would somehow tune the fluorescence properties of the related complexes as HS^−^ sensors.

## 2. Experimental Section

### 2.1. Materials

Chemicals used for the synthetic work were obtained from Sigma-Aldrich or Strem Chemicals and were of reagent grade. They were used without further purification. NaSH (Alfa Aesar) in an aqueous solution was used as HS^−^ source to the end concentration specified in the figure captions. Complex **3** was synthesized by following literature procedures [14]. As for the synthesis of the other complexes prepared in this work, 4-(diethylamino)salicylaldehyde was substituted with the appropriate salicylaldehyde derivative as detailed below.

### 2.2. General

High resolution (HR) MALDI and electron spray ionization (ESI) mass spectra were performed by using a Bruker solariX XR Fourier transform ion cyclotron resonance (FT-ICR) mass spectrometer (Bruker Daltonik GmbH, Bremen, Germany) with a 7 T refrigerated actively-shielded superconducting magnet (Bruker Biospin, Wissembourg, France). The samples were ionized in positive ion mode using either the MALDI ion or the ESI ion source. The mass range was set to *m*/*z* 150–2000. The laser power was set to 15% and 15 laser shots were used for each measurement. Mass spectra were calibrated externally by a mix of peptide clusters in MALDI ionization positive ion mode. As for the ESI mass spectra, they were calibrated externally by using NaTFA in solution in negative ion mode. A linear calibration was applied.

NMR spectra were recorded on a Bruker AVANCE 400 NMR instrument (^1^H NMR, 400.13 MHz; ^13^C NMR, 100.62 MHz) or on a 600 MHz spectrometer [600 (^1^H NMR) and 151 MHz (^13^C NMR)] using 5 mm o.d. NMR tubes. The chemical shifts were reported in δ (ppm) referenced to SiMe_4_. Typically, 5 mg of the complex in 0.5 mL of the deuterated solvent were used for each experiment. MilliQ water is the water filtered with a Millipore filter apparatus.

**Synthesis and characterization of complex 1**. A mixture of salicylaldehyde (0.108 mL, 1 mmol) and thiophene-3,4-diamine (0.06 g, 0.5 mmol) in 100 mL of absolute ethanol was stirred for 1 h at room temperature. Then, 1 eq of Zn(CH_3_COO)_2_*2H_2_O (0.114 g, 0.5 mmol) was added and the mixture was stirred for an additional 16 h at room temperature. A yellow solid was recovered by filtration on common filter paper, this was then washed with cold methanol and dried under vacuum (yield 75 %).

^1^H NMR [400 MHz, DMSO-*d_6_*]: *δ* = 9.01 (s, 2H, C***H*** = N), 7.86 (s, 2H, ***H*** thiophene), 7.32–7.20 (m, 4H, ***H*** aromatic), 6.72 (d, 2H, ***H*** aromatic), 6.54 (t, 2H, ***H*** aromatic). ^13^C NMR [151 MHz, DMSO-*d_6_*]: *δ* = 172.4 (2C), 163.6 (2CH), 141.0 (2C), 135.6 (2CH), 134.0 (2CH), 123.2 (2CH), 119.3 (2C), 113.1 (2CH), 108.9 (2CH). Emission (DMSO, λ_exc_ = 402 nm), λ_max_, nm (quantum yield,Φ_F_): 497 nm (0.01).

**Synthesis and characterization of complex 2:** A mixture of *o*-vanillin (0.158 g, 1 mmol) and thiophene-3,4-diamine (0.06 g, 0.5 mmol) in 100 mL of absolute ethanol was stirred for 1 h at room temperature. Then 1 eq of Zn(CH_3_COO)_2_*2H_2_O (0.114 g, 0.5 mmol) was added and the mixture was left under stirring for an additional 16 h at room temperature. A yellow solid was recovered by filtration on common filter paper; this was then washed with cold methanol and dried under vacuum (yield 65 %).

^1^H NMR [400 MHz, DMSO-*d_6_*]: *δ* = 8.99 (s, 2H, C***H*** = N), 7.84 (s, 2H, ***H*** thiophene), 6.93–6.84 (dd, 4H, ***H*** aromatic), 6.48 (t, 2H, ***H*** aromatic), 3.78 (s, 3H, OC***H***_3_). Due to the scarce solubility of complex **2** in DMSO-d6, the authors could not register the ^13^C NMR spectrum.

**Synthesis and characterization of complex 3**: A mixture of 4-(diethylamino)salicylaldehyde (0.200 g, 1 mmol) and thiophene-3,4-diamine (0.06 g, 0.5 mmol) in 100 mL of absolute ethanol was stirred for 1 h at room temperature. Then 1 eq of Zn(CH_3_COO)_2_*2H_2_O (0.114 g, 0.5 mmol) was added and the mixture was left under stirring for an additional 16 h at room temperature. A green solid was recovered by filtration on common filter paper; this was then washed with cold methanol and dried under vacuum (yield 55 %).

^1^H NMR [400 MHz, DMSO-*d_6_*]: *δ* =8.63 (s, 2H, C***H*** = N), 7.50 (s, 2H, ***H*** thiophene), 7.04 (d, 2H, ***H*** aromatic) 6.07 (d, 2H, ***H*** aromatic), 5.81 (s, 2H, ***H*** aromatic), 3.40 (8H, NC***H***_2_CH_3_), 1.14 (t, 12H, NCH_2_C***H***_3_). ^13^C NMR [151 MHz, DMSO-*d_6_*]: *δ* = 173.8 (2C), 160.1 (2CH), 152.3 (2C), 141.8 (2C), 136.9 (2CH), 111.0 (2C), 104.9 (2CH), 101.4 (2CH), 101.3 (2CH), 43.8 (2*C*H_2_), 12.9 (2*C*H_2_).

**Synthesis and characterization of complex 4:** A mixture of 2,4-dihydroxybenzaldehyde (0.144 g, 1 mmol) and thiophene-3,4-diamine (0.06 g, 0.5 mmol) in 100 mL of absolute ethanol was stirred for 1 h at room temperature. Then 1 eq of Zn(CH_3_COO)_2_*2H_2_O (0.114 g, 0.5 mmol) was added and the mixture was stirred for an additional 16 h at room temperature. A brown solid was recovered by filtration on common filter paper; this was then washed with cold methanol and dried under vacuum (yield 50 %).

^1^H NMR [400 MHz, DMSO-*d_6_*]: *δ* = 9.73 (s, 2H, O***H***), 8.79 (s, 2H, C***H*** = N), 7.63 (s, 2H, ***H*** thiophene), 7.12 (d, 2H, ***H*** aromatic) 6.05 (s, 4H, ***H*** aromatic). ^13^C NMR [151 MHz, DMSO-*d_6_*]: *δ* = 174.5 (2C), 163.5 (2C), 161.8 (2CH), 141.3 (2C), 137.3 (2CH), 113.6 (2C), 106.7 (2CH), 106.6 (2CH), 104.7 (2CH).

**Synthesis and characterization of complex 5:** A mixture of 3,5-dibromosalycilaldehyde (0.291 g, 1 mmol) and thiophene-3,4-diamine (0.06 g, 0.5 mmol) in 100 mL of absolute ethanol was stirred for 1 h at room temperature. Then 1 eq of Zn(CH_3_COO)_2_*2H_2_O (0.114 g, 0.5 mmol) was added and the mixture was stirred for an additional 16 h at room temperature. A brown solid was recovered by filtration on common filter paper; this was then washed with cold methanol and dried under vacuum (yield 50 %).

^1^H NMR [400 MHz, DMSO-*d_6_*]: *δ* = 8.97 (s, 2H, C***H*** = N), 7.83 (s, 2H, ***H*** thiophene), 7.74 (s, 2H, ***H*** aromatic), 7.55 (s, 2H, ***H*** aromatic). ^13^C NMR [151 MHz, DMSO-*d_6_*]: *δ* = 165.1 (2C), 162.1 (2CH), 140.7 (2C), 137.3 (2CH), 136.8 (2CH), 121.1 (2C), 118.8 (2CH), 110.4 (2C), 101.6 (2C). Emission (DMSO, λ_exc_ = 410 nm), λ_max_, nm (quantum yield, Φ_F_): 506 nm (0.07).

**Absorbance and fluorescence measurements.** Absorption spectra were recorded on a Cary-50 Spectrophotometer using a 1 cm quartz cuvette (Hellma Benelux bv, Rijswijk, Netherlands) and a slit-width equivalent to a bandwidth of 5 nm. Fluorescence spectra were measured on a Cary Eclipse Spectrophotometer in a 10 × 10 mm^2^ airtight quartz fluorescence cuvette (Hellma Benelux bv, Rijswijk, Netherlands) with an emission band-pass of 10 nm and an excitation band-pass of 5 nm. Both absorption and fluorescence measurements were performed in DMSO solutions at 25 °C. Fluorescence emission spectra were registered by exciting the samples at a specific wavelength (as stated in the figure captions). 

Fluorescence quantum yield (Φ_F_) values were measured in optically diluted solutions using the commercial dye Cy3 NHS (Φ_F_ = 0.15 in MilliQ water) as standards, according to the equation [15]:Φ_F_^s^ = Φ_F_^r^ (*I*_s_/*I*_r_)(*A*_r_/*A*_s_)(*η*_s_/*η*_r_)^2^
where indexes s and r denote the sample and reference, respectively. *I* stands for the integrated emission intensity, *A* is the absorbance at the excitation wavelength, and *η* is the refractive index of the solvent. The optical density of complexes **1** and **5** and standards were kept below 0.1. The uncertainty in the determination of Φ_F_ is ±15%.

The HS^−^ titration experiments were performed as follows: the cuvette was filled with sample solutions in DMSO. Then µL amounts of HS^−^ solutions in MilliQ water (to the end concentrations specified in the figure captions) were injected via gas-tight syringe at intervals of 1 min between subsequent additions. The experiment ended when no changes in the fluorescence intensities could be detected upon HS^−^ addition.

**NMR characterization of the complexes 1–5 upon addition of HS^−^.** The NMR tube was charged with the free complex solutions in DMSO-*d_6_*, then NaSH solid (to the end concentrations specified in the figure captions) was added and the spectra registered. 

**Preparation of the paper-based strip sensors.** Filter paper sheets were cut into 1 cm^2^ regular strips, which were subsequently soaked in DMSO solutions of complex 5 (10 × 10^−3^ M) for about 24 h at 37 °C. Then the filter strips were removed from the solution and dried in air for about 12 h at room temperature. Ten µl of NaSH solutions in MilliQ water and cell culture medium (at the concentrations specified in the figure captions) and cell-conditioned media were dropped directly on the filter paper strips. Once dry, the filter paper strips were analyzed under an ultraviolet lamp (Spectroline ENF-240C/FE) working at 365 nm wavelength irradiation.

**Cell culture and cell-conditioned media preparation.** HepG2 cells (Human hepatocellular liver carcinoma cell line) were grown in Minimum Essential Medium (MEM) supplemented with 10% fetal bovine serum (FBS), 2 mM Glutamine, 1 mM non-essential amino acids, and 1% antibiotics (penicillin/streptomycin, 100 U/mL). Cells were maintained in a humidified incubator at 37 °C in 5% CO_2_/95% air. To prepare cell-conditioned media, about 5 × 10^5^ cells were seeded in each well of a 6-well plate and cultured with 1 mL of cell culture medium for 24 h. To induce endogenous production of HS^−^, cells were treated with 800 µM of H_2_O_2_ for 1 h. After the incubation, conditioned media were collected and used for testing paper-based strip sensors as described above.

**Computational details.** All electronic computations were performed at the density functional level of theory using the range separated hybrid functional CAM-B3LYP with TZVP basis set as implemented in the Gaussian package (G16) [16]. That combination of the functional and basis set waschosen because it leads to accurate predictions, as discussed in previous works [17,18,19,20]. Time dependent DFT (TD-DFT) has been employed for treating all excited states. Spin-orbit coupling elements have been computed by PySOC code [21]. Effects due to solvent polarization were included by the polarizable continuum model (PCM) [22].

## 3. Results and Discussion

Complexes **1**–**5** were synthesized from thiophene-3,4-diamine and the proper substituted salicylaldehyde by following literature procedures [14], as detailed in the experimental part of the present work. Complexes were characterized by high resolution ESI mass spectrometry (Appendix A), ^1^H NMR (Appendix A), and ^13^C analysis (Appendix A).

### 3.1. HS^−^ Interaction with Complexes ***1***–***5***

As largely documented in the literature, when HS^−^ interacts with a metal complex, three scenarios are possible: (i) displacement of the metal from the ligand to generally produce fluorescence changes via HS^−^-mediated precipitation of metal sulfides, (ii) binding of HS^−^ to the metal center, and (iii) HS^−^-mediated reduction of the metal center (in the case of redox active metals).

^1^H NMR is a very powerful technique to assess which situation occurs when HS^−^ interacts with a target complex. Examples of zinc complexes that result in scenario (i) upon interaction with HS^−^ are well documented in the literature [23]; in these cases, one of the most striking pieces of evidence, usually found in the ^1^H NMR spectrum of the reaction mixture between the target complex and HS^−^, is the restoration of the spectrum of the starting ligand. Different appearances of a high field resonance at δ ~ −2.3 ppm constitute as strong evidence of HS^−^ binding to the zinc center. Drawing upon these considerations, to obtain indications on the mechanism by which HS^−^ interacts with complexes **1**–**5**, the authors examined the reactions by NMR spectroscopy, considering that the scenario (iii) can be discarded since zinc is a non-redox metal.

When adding NaSH to the DMSO-*d*_6_ solutions of complex **1**–**5**, a shift of the proton resonances and the appearance of a high field resonance at δ ~ −2.95 ppm (Appendix A) for all the complexes under investigation occurred, thus pointing to the coordination of the SH group to the zinc centers. The only exception is represented by complex **4,** for which the high field resonance is missing (Appendix A) (e.g., the broad band around −3.5 ppm is that of the free HS^−^, as reported in the literature in similar experimental conditions) [24]. The latter finding is most likely due to an exchange of the SH with the OH groups on the ligand mediated by the trace amounts of water in the deuterated DMSO [25].

The UV-vis spectra of complexes **1**–**5** before and after addition of an excess of HS^−^ are reported in Figure 1.

Based on theoretical computations, *vide infra*, the lowest energy electronic transitions of **1** and **5** are ligand to ligand transitions, without any metal contribution. For the HS^−^ complexes, all transitions involve orbitals that are distributed over the ligand and the HS^−^ anion. As shown in Figure 1, for all the title complexes the interaction with HS^−^ results in a change of the initial UV-visible spectrum, which points to the formation of a new species, confirms the results drawn by NMR experiments.

In a subsequent instance, the authors studied the fluorescence response of complexes **1**–**5** before and after treatment with HS^−^ (Figure 2).

The fluorescence screening indicates a fluorescence quenching for complexes **1**–**4** with the only exception of complex **5**, which undergoes a fluorescence enhancement in the presence of an excess of HS^−^. Therefore, complex **5** results are the most promising construct for the implementation of a real sensor (e.g., systems harnessing fluorescence enhancement or “turn on” as a result of the recognition event are inherently more sensitive than “turn-off” sensors that exploit fluorescence quenching upon analyte binding).

As mentioned above, the authors aim to explore whether different combinations of groups on the ligand structure (e.g., bridging diamine and substituents on the salicylaldehyde units with different electron-withdrawing and electron-donating abilities) would tune the fluorescence properties of the related complexes as HS^−^ sensors. The results displayed in Figure 2 suggest that passing from a diaminomalonitrile (DAMN) [11] to a thiophene bridge plays a key role on the fluorescence response of the different systems. More specifically, while in the case of the complexes with the DAMN bridge, only the complex obtained by N-diethylsalicylaldehyde experiences a fluorescence quenching upon HS^−^ coordination to the zinc center [11] for the thiophene-bridged complexes presented here; even in the presence of the same substituents on the phenolate groups, the fluorescence is quenched when HS^−^ binds to the zinc center.

### 3.2. Computational Study

In order to further investigate the photophysical properties of compounds **1**–**5** and their adducts with HS^−^, and to understand the reasons of the different fluorescence behavior with respect to the sister compounds bearing a DAMN bridging unit, discussed in the authors’ previous publication [11], the authors performed a computational analysis at the time dependent density functional theory (TD-DFT) level. The authors focused on complexes **1** and **5**, which are representative of the whole class. Minimum energy geometries of **1** and **5** and of their HS^−^ adducts (also considering the possibility of multiple adducts) were computed both for the ground state and for the first excited singlet states. Figure 3 displays the computed ground state optimum geometries of **1** and its HS^−^ complexes (a similar geometry has been also found for **5** and its HS^−^ complex, see Appendix A).

Complexes **1** and **5** exhibit the square planar nuclear configuration observed for Zn complexes (C_2v_ point group), with the metal atom in the plane of the ligand (see Figure 3). Upon coordination of a single HS^−^, the metal ion is slightly displaced out of the ligand plane and symmetry is lost. The formation of the single adduct is predicted to be exoergonic (Δ*E* = −0.77 eV for complex **1**, Δ*E* = −0.90 eV for complex **5**), whereas the double adduct is not predicted to be a stable species, as confirmed by DFT computation where the second HS^−^ is moved away from the metal center during geometry optimization.

In the first excited singlet state (S_1_), the geometry of both **1** and **5** and of their HS^−^ adducts are only slightly distorted with respect to the ground state (S_0_). Emission from S_1_ is predicted to be an electric dipole allowed both for **1** and **5** and for their HS^−^ adducts. Computed vertical and adiabatic excitation energies are reported in Table 1, together with the oscillator strengths for the S_1_ ← S_0_ transitions.

For **1**, 3 electric dipole allowed transitions are predicted in the spectral range 300–500 nm, 2 of which are very close, so that they are not distinguished in the experimental spectrum, which shows only 2 peaks (see Figure 1). However, a meaningful comparison between predicted and observed absorption spectra would require a band shape simulation, with the computations of Franck–Condon integrals [26], which is far beyond the qualitative purposes of the present computational analysis.

Since all the species emissions from S_1_ are electric dipole allowed transitions, the different behavior observed for **1** and **5** and for their HS^−^ complexes must be related with the possible existence of non-radiative decay paths. The authors thus investigated the energy location of the lowest triplet states, which could be responsible for the different fluorescence quantum yields of **1** and **5** and their HS^−^ adducts (see Appendix A).

The energies of the four lowest triplet states are reported in Figure 4; T5 always lies above in energy compared to S_1_, and therefore it should not be involved in non-radiative decay paths. T_2_ and T_1_ are significantly lower in energy than S_1_ for all the species, and therefore, based on the energy gap rule, the direct transition S_1_ → T_1_ and S_1_ → T_2_ should not be an efficient decay path. The triplet states closer in energy to S_1_ are T_3_ and T_4_, whose spin-orbit couplings are reported in Table 2.

Concerning complex **1**, the exoergonic S_1_ → T_3_ transition is slightly more favored after HS^−^ coordination, as demonstrated by the higher spin-orbit coupling (SOC) and the lower energy difference between the electronic states. Moreover, S_1_ → T_4_ transition, though endoergonic before and after the coordination, is much more favored in the latter case since the electronic states involved become almost isoenergetic and the spin orbit coupling is more than 10 times higher. Thus, a strong quenching of fluorescence is to be expected in the presence of HS^−^, as is observed (see Figure 2).

Concerning **5**, on the contrary, the spin-orbit coupling for the transition S_1_ → T_3_ becomes smaller, and the energy difference between S_1_ and T_4_ energies is much higher after the coordination of HS^−^. Both effects concur in making the non-radiative decay paths via triplet states significantly slower, thus an enhancement in fluorescence is expected after the adduct is formed, as experimentally observed in Figure 2.

### 3.3. Sensing Applications

In the following experiments, the authors focused on complex **5**, which was the only one of the series undergoing a fluorescence enhancement in the presence of HS^−^ and thus useful for practical sensing measurements, as discussed above.

To assess whether the modification of fluorescence emission of complex **5** depends on the amount of HS^−^, the authors monitored the change of its initial fluorescence intensity after the addition of increasing concentrations of NaSH. Figure 5A shows the response of the fluorescence switching of complex **5** for a series of subsequent measurements with increased concentrations of NaSH. Figure 5B displays the fluorescence intensity values at 490 nm fit against HS^−^ concentration, which can be used as a calibration of the system. In a titration experiment, one expects the titration curve to become horizontal when the endpoint of the titration is reached. In the present case, the start of the bending occurs at around 50 µM.

As a next step, the authors also explored the chromogenic capability of the complexes under investigation for the detection of HS^−^. In the presence of HS^−^, a color change visible to the naked eye occurred for complex **5** (see Appendix A) when dissolving the complex in DMSO. To assess the response time of complex **5** in the detection of HS^−^, the fluorescence intensity of complex **5** was monitored as a function of time upon addition of an excess of HS^−^ (see Appendix A). Further, to determine the selectivity of complex **5** in the recognition of HS^−^, the authors checked its fluorescence intensity in the presence of biologically relevant and potentially competing thiols (e.g., L-cysteine (L-cys) and glutathione (GSH), see Appendix A). Different than that observed in the presence of HS^−^, the fluorescence intensity of complex **5** exhibits moderate quenching both with L-cys and with GSH (see Appendix A).

Encouraged by these results and as a further practical application, the authors tested whether complex **5** could function as a paper-strip based sensor of HS^−^. Figure 6 displays the outcome of an experiment under UV light (at 365 nm wavelength irradiation) in which it is evident that, as long as the authors add increasing amounts of HS^−^, the blue paper strip with complex **5** (left picture) turns pink (middle and right pictures).

To test if this finding could have any application for biological purposes, the authors repeated this same test by dissolving HS^−^ in a cell culture medium. As expected, a first color change of the paper strips was observed after the addition of medium only (“Medium” in Figure 7), compared to the original complex **5**-loaded paper strip (“Free” in Figure 7), likely due to the presence of some colored components in the culture medium. More interestingly, an evident chromatic variation was detected with increasing amounts of HS^−^ (Figure 7 and Appendix A). HS^−^ concentrations chosen for the assay, 0.001, 0.01, and 0.1 mg/ml, corresponded to 17.8, 178, and 1780 µM, respectively, thus falling into physiological range of HS^−^, as reported in the literature [27,28].

Therefore, to test the detection capability of complex **5** in more physiological conditions, the authors used culture media previously incubated with HepG2 cells for 24 h at 37 °C in order to assess basal levels of endogenous HS^−^. Different chromatic variations of complex **5**-loaded paper strips were very evident under UV light (Figure 8 and Appendix A). In particular, the strip in contact with medium conditioned by the cells (Figure 8b) assumed a color more similar to samples treated with lower concentrations of NaSH (0.001 mg/mL) (Figure 7) than non-conditioned medium (Figure 7 and Figure 8a). Furthermore, the authors tested the capability of complex **5**-loaded paper strips to detect variations of endogenous HS^−^ levels in cell-conditioned media. To this aim, HepG2 cells were treated for 1 h with 800 µM H_2_O_2_ to induce an increment in HS^−^ production [29]. Figure 8d shows that the chromatic change occurring with conditioned medium upon cell treatment with H_2_O_2_ was similar to a paper-strip put in contact with higher concentrations of NaSH (0.1 mg/mL) (Figure 7). Conversely, a slight color variation was observed by adding non-conditioned medium +800 µM H_2_O_2_ to complex **5**-loaded paper strips, used as a control (Figure 8c). These preliminary observations of basal and stress-induced levels of endogenous HS^−^ revealed by complex **5** are in general agreement with other detection systems used with cell lines and were previously reported in literature [30]. Taken altogether, these findings suggest a promising application potential of this system as a chemosensor of HS^−^ in complex biological fluids.

## 4. Conclusions

In this work, several pieces of evidence are presented that demonstrate how the complexes under investigation can successfully function as H_2_S sensors by harnessing a ‘coordinative-based’ approach [31]. A computational analysis at the time dependent density functional theory (TD-DFT) level was performed both to investigate the different photophysical features observed for compounds **1**–**5** and their adducts with HS^−^ and to understand the reasons of the different fluorescence behavior with respect to the sister compounds bearing a DAMN bridging unit [11]. The results presented here highlight that the electronic state of the diamine moiety strongly modulates the photophysical properties of the salen complexes, as well as their fluorescence response to the target analyte. The DFT investigation suggests that the different fluorescence behaviors observed for **1** and **5** upon interaction with HS^−^ must be related to the HS^−^ ability to change non-radiative decay paths, rather than dipole strengths. The first screening with complex **5**-loaded paper strips in cell-conditioned culture medium indicated the potential of this family of complexes to function as chemosensors of HS^−^ in complex biological fluids.

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
