# Peer review of "Paper-Strip-Based Sensors for H2S Detection: A Proof-of-Principle Study"

_sensors, 2022, doi:10.3390/s22093173_

Round 1
Reviewer 1 Report
The authors synthesized and characterized 5 different Zn-organic frameworks and investigated their properties as HS- indicators in liquid medium. A complete spectroscopic investigation has been performed as well as a modelization of the ground and excited states of selected complexes.
Proof of concept color indicators on paper strips have been realized and their color shift is characterized upon exposure to different concentrations of HS-.
I found their work complete and clearly presented, with the SI quite useful and strongly supporting the manuscript.
I suggest its publication in present form.
Author Response
REVIEWER 1
The authors synthesized and characterized 5 different Zn-organic frameworks and investigated their properties as HS- indicators in liquid medium. A complete spectroscopic investigation has been performed as well as a modelization of the ground and excited states of selected complexes.
Proof of concept color indicators on paper strips have been realized and their color shift is characterized upon exposure to different concentrations of HS-.
I found their work complete and clearly presented, with the SI quite useful and strongly supporting the manuscript.
I suggest its publication in present form.
Thank you very much for reviewing our manuscript and for your positive comment.

Reviewer 2 Report
The article “Paper-strip-based sensors for H2S detection: a proof-of-principle study” is dealing with the preparation of fluorescent zinc complexes to study their potential as H2S sensors. Zn complexes were characterized by 1H and 13C NMR and high resolution ESI mass spectrometry. Interaction between HS- and Zn complexes were studied using 1HNMR and UV-Vis and fluorescence measurements and the photophysical properties were investigated by computation (TD-DFT). Finally detection capability of the best performed Zn complex as chemosensor of HS- in complex biological fluids was studied using a screening procedure.
The paper is well-written and documented. However there are still some spelling errors to be corrected and information especially concerning NMR spectroscopy which should be included. My recommendation is minor revision.
Comments and questions:
- Introduction
p.1: In spite of the discovery that it is an endogenously-produced biomolecule” instead of “In spite of the discovery that is an endogenously-produced biomolecule”
p.1: “gasotransmitters” instead of “gasotrasmitters”
p.1: “fluorescence” instead of “fuorescence”
p.1: “salicylaldehyde units” instead of “salycilaldehyde units”
- Experimental section
p.3: : “salicylaldehyde derivative” instead of “salycilaldehyde derivative”
p.3: “for each measurement” instead of “each measure”
p.3: which filter was used for solid recovery by filtration? Please comment.
- Results and Discussion
p.5: why 13C spectra were not included in the supporting information? Please comment.
p.5: why in figure S15 the δ scale is not the same as for Figure S12, S13, S14 and S16? The resonance signals of phenolic OH are not found. Please comment. It is not clear to me if the complex as drawn in Figure S15 is present and SH should be replaced by OH? Please comment.
p.6: “free HS-“ instead of “HS- free”
p.8: “N-diethylsalicylaldehyde” instead of “N-diethylsalycilaldehyde”
p.9: “ = −0.90 eV for complex 5” instead of “ΔE = −0.90 eVfor complex 5”
p.9: “two of which are very close” instead of “two of which very close”
p.10: “which could be responsible for the different” instead of "which could be responsible of the different“
p.11: “for complex 1 and 5” instead of “for complex 1and 5”
p.11: “T2 and T1 are” instead of “T2 and T1are”
p.11: “S1→ T4 transition” instead of “S1→ T4transition”
p.11: “spin-orbit coupling (SOC)” instead of “spin-orbit coupling”
p.13: “under the UV light”: which wavelength (range) was used?
Reviewer 3 Report
Comments and Suggestions for Authors
This article presented a principle study on paper strip-based sensors for H2S detection. In this study, the authors explained the interaction of a suite of fluorescent zinc complexes with H2S. However, there are some issues to be clarified in this study.
Comment 1. Although the primary purpose of this paper is the proof of principle study, I think that the author should confirm at least the basic properties of Complex 5 as H2S sensors, such as detection limit, detection time, and selectivity. Please elaborate on the issues mentioned above.
Comment 2. There are some typos in the manuscript. Please revise them. Introduction: 'exhert' can be modified to 'exert.' 'ZnSalen complexes' can be modified to 'Zn-Salen complexes.' Please list the full name of the analyses before using abbreviations unfamiliar to the general readers, such as HR MALDI and ESI mass spectra.
Author Response
REVIEWER 3
This article presented a principle study on paper strip-based sensors for H2S detection. In this study, the authors explained the interaction of a suite of fluorescent zinc complexes with H2S. However, there are some issues to be clarified in this study.
Thank you very much for reviewing our manuscript. A point to point reply to your comments follows:
Comment 1. Although the primary purpose of this paper is the proof of principle study, I think that the author should confirm at least the basic properties of Complex 5 as H2S sensors, such as detection limit, detection time, and selectivity. Please elaborate on the issues mentioned above.
Following the suggestions of the reviewer we performed additional experiments in the revised version of our manuscript to assess selectivity and response time of complex 5 in the detection of H2S. We added the outcome of these new experiments in figs S23 and S4.
Comment 2. There are some typos in the manuscript. Please revise them. Introduction: 'exhert' can be modified to 'exert.' 'ZnSalen complexes' can be modified to 'Zn-Salen complexes.' Please list the full name of the analyses before using abbreviations unfamiliar to the general readers, such as HR MALDI and ESI mass spectra.
As suggested by the reviewer we went throughout the all manuscript and revised all the typos, furthermore we also listed the full name of all the abbreviations used.

Round 2
Reviewer 3 Report
I would recommend that the manuscript accepts in its present form.